# Current Clinical Trials in Traumatic Brain Injury

**DOI:** 10.3390/brainsci12050527

**Published:** 2022-04-21

**Authors:** Zubair Ahmed

**Affiliations:** 1Institute of Inflammation and Ageing, College of Medical and Dental Sciences, University of Birmingham, Edgbaston, Birmingham B15 2TT, UK; z.ahmed.1@bham.ac.uk; 2Centre for Trauma Sciences Research, University of Birmingham, Edgbaston, Birmingham B15 2TT, UK; 3Surgical Reconstruction and Microbiology Research Centre, National Institute for Health Research, Queen Elizabeth Hospital, Birmingham B15 2TH, UK

**Keywords:** traumatic brain injury, clinical trials, tranexamic acid, fibrinolysis, efficacy

## Abstract

Traumatic brain injury (TBI) is one of the leading causes of morbidity, disability and mortality across all age groups globally. Currently, only palliative treatments exist, but these are suboptimal and do little to combat the progressive damage to the brain that occurs after a TBI. However, multiple experimental treatments are currently available that target the primary and secondary biochemical and cellular changes that occur after a TBI. Some of these drugs have progressed to clinical trials and are currently being evaluated for their therapeutic benefits in TBI patients. The aim of this study was to identify which drugs are currently being evaluated in clinical trials for TBI. A search of ClinicalTrials.gov was performed on 3 December 2021 and all clinical trials that mentioned “TBI” OR “traumatic brain injury” AND “drug” were searched, revealing 362 registered trials. Of the trials, 46 were excluded due to the drug not being mentioned, leaving 138 that were completed and 116 that were withdrawn. Although the studies included 267,298 TBI patients, the average number of patients per study was 865 with a range of 5–200,000. Of the completed studies, 125 different drugs were tested in TBI patients but only 7 drugs were used in more than three studies, including amantadine, botulinum toxin A and tranexamic acid (TXA). However, previous clinical studies using these seven drugs showed variable results. The current study concludes that clinical trials in TBI have to be carefully conducted so as to reduce variability across studies, since the severity of TBI and timing of therapeutic interventions were key aspects of trial success.

## 1. Introduction

Traumatic brain injury (TBI) is one of the leading causes of morbidity, disability and mortality across all age groups [1,2]. The biggest causes of TBI are motor vehicle accidents, falls and interpersonal violence. More than 50 million individuals suffer a TBI each year and there are approximately 3.2 million TBI survivors who experience post-TBI complications, which include neurological and psychological problems as well as long-term disability [3,4,5]. In the UK alone, clinical management of TBI costs the economy GBP 15 billion per year, representing a significant burden on the UK economy.

TBI is classified into different categories including closed head, penetrating and explosive blast TBI. TBI patients suffer a variety of symptoms including headache, nausea, seizures, amnesia, aggression and anxiety. These symptoms can appear within seconds after a TBI, and some of the effects can last months to years [1,6]. Closed head TBI typically presents after blunt trauma impact mainly from motor vehicle accidents, falls and sports. Penetrating TBI occurs when a foreign body penetrates the skull and traverses through into the brain parenchyma. Explosive blast TBI occurs primarily in war-related TBI, where the rapid shock waves of a blast transmit kinetic energy from the skull and into the parenchyma causing deformation of the brain. All TBIs result in damage to the brain tissues, including the vasculature, diffuse axonal injury, compromised blood brain barrier and cerebral oedema.

Damage to the brain occurs both from the primary and secondary injury processes. The primary injury results from the direct mechanical forces acting on the brain during the initial insult whilst the secondary injury results from tissue and cell damages that follow the initial insult. The primary injury causes both focal and diffuse consequences which leads to tissue necrosis, hematomas and intracerebral hemorrhages, and eventually damages axons, oligodendrocytes and the vasculature, leading to oedema and ischemic brain damage [7,8,9,10]. The secondary brain injury occurs as a result of the biochemical, cellular and physiological changes that follow a TBI. For example, excitotoxicity, mitochondrial dysfunction, oxidative stress, lipid peroxidation, neuroinflammation, axon degeneration and apoptosis all occur after a TBI and contribute to the secondary injury processes [11]. 

Many of the treatment regimens for TBI focus on stabilization of the injured brain and prevention of secondary injury processes. Since secondary injury characteristics develop progressively, a therapeutic window of opportunity exists in order to protect further loss of neurons and glia as well as targeting excitotoxicity, inflammation, oxidative stress and apoptosis. An increased understanding of the clinical characteristics and the underlying complex pathophysiological mechanisms of TBI has led to the development of several novel and promising therapeutic approaches that have shown positive effects in preclinical studies. Many of these drugs have progressed from preclinical studies into clinical evaluation of their usefulness in treating TBI. 

Some of the long-term consequences of TBI include problems with balance, motricity, headaches, cognitive and behavioral problems and post-traumatic epilepsy [12,13,14,15]. Many patients also suffer from anxiety or depression, and management of these symptoms are top priorities among the most frequently perceived needs [16,17,18]. About a third of TBI subjects were lower moderate to severely disabled, requiring help with daily living activities, whilst a third were at an upper moderate disability level, meaning that they were independent inside and outside their homes but had reduced work capacity, social, leisure and family or friendship disruption [16,19,20]. Counteracting these long-term consequences of TBI must be factored into future TBI clinical trials. 

The aim of this review is to present an overview of the clinical trials that were registered with ClinicalTrials.gov on 3 December 2021 and present data on the number of trials that have been completed and the number of patients in each trial, as well as reviewing the drug treatments that are being tested. The review also aims to provide some guidelines on improving future clinical trial design, which need to be considered in this area of unmet medical need.

## 2. Trials Registered in ClinicalTrials

The ClinicalTrials.gov (accessed on 3 December 2021) database was searched with the terms “TBI” OR “traumatic brain injury” AND “drug”. As of 3 December 2021, there were 361 trials registered on Clinical Trials.gov that were returned after searching for “TBI” OR “traumatic brain injury” AND “drug” (Figure 1). Of all studies retrieved, 46 were excluded or not relevant due to the drug not being mentioned. Of the trials listed, 138 were completed, 116 were withdrawn, suspended or the status was unknown, 46 were still recruiting, 5 were active but not yet recruiting, 13 were not recruiting and 2 were enrolling by invitation (Figure 1). 

Although all of the studies included 267,298 patients, the average number of patients was 865 per study with a range of 5–200,000 and a median of 50 patients/study.

### 2.1. Completed Studies

Of the studies completed, the highest numbers were completed between 2016 (16) and 2017 (17), with fewer studies completed since then (Figure 2). The total number of patients in these studies was 40,947, with an average of 1742 patients/study, a range of 10–14,959 and a median of 130 patients. Patients were stratified into 46 different age groups in the 138 completed studies, ranging from categories such as child, adult, older adult and anywhere between 2 to 100 years old (Figure 3). The largest age groups for enrollment into a TBI trial were 18 years and older (3600 participants), 18 to 60 years old (13,043 participants) and 18 to 65 (14,959 participants) (Figure 3).

Of the completed studies, 24 were early Phase I, 8 were Phase I/II, 35 were Phase II, 5 were Phase II/III, 20 were Phase III, 22 were Phase IV and 24 were not applicable to be assigned to a phase of study (Figure 4). 

Between the completed studies, 125 different drugs for TBI were evaluated. However, only seven drugs were used in three or more studies across the different clinical phases of study, including amantadine, botulinum toxin type A, hyperbaric oxygen, methylphenidate, NNZ-2566, Rivastigmine and tranexamic acid (TXA) (Figure 5). 

### 2.2. Phase IV Studies

There are currently 18 drugs in Phase IV clinical trials. These are drugs that have been approved previously but are being investigated in studies where the side effects caused over time are being evaluated. These include Dipeptiven, propranolol, methylphenidate, rHGF, Lisdexamfetamine, Venlafaxine, Rivastigmine, citalopram (celexa), Doxycycline, simvastatin, Enoxaparin, Levetiracetam, Androgel (testosterone), amantadine, botulinum toxin type A, intravenous acetaminophen, Nuedexta and 20% mannitol.

## 3. Characteristics of Completed Studies

### 3.1. Completed Studies with Results

There were only 44 clinical trials in TBI that have been completed and where results are available. Table 1 provides a summary of these studies, their primary outcomes and the results posted on ClinicalTrials.gov (accessed on 3 December 2021). Surprisingly, only six study results have been published (NCT00313716, NCT01322048, NCT02012582, NCT01750268, NCT02270736, NCT01463033). Of the 44 studies, 30 showed no statistical difference between treatment and controls, 7 studies showed better results in the treatment arm but lacked statistical data, 4 studies had only a single arm and so comparisons could not be made, 2 studies showed worse outcomes in the treatment arm, whilst only 1 study demonstrated significantly better results in the treatment arm (NCT02270736). Twenty-four studies failed to meet their recruitment targets whilst seventeen studies recruited more patients than posted on ClinicalTrials.gov (accessed on 3 December 2021).

### 3.2. Summary of Evidence for Drugs Appearing in Three or More TBI Studies

#### 3.2.1. Amantadine

Amantadine is a dopaminergic agent and is an antagonist of N-methyl-D-aspartate (NMDA), approved by the FDA for use in the prevention of influenza and Parkinson’s disease [21]. Amantadine is most commonly prescribed for patients with disorders of consciousness and who are undergoing inpatient neurorehabilitation, although its mechanism of action is unclear. However, several clinical trials have shown positive effects with amantadine in terms of neurobehavioral recovery, cognitive function and improved disability rating scores [22,23,24]. Other studies have either shown no benefits of amantadine or some positive benefits; however, these studies only had 20–25 participants [25,26]. In a study in 2018, the use of amantadine to improve cognitive performance was not supported in a multi-site, randomized controlled trial (RCT) [27]. Although amantadine is well tolerated, definite improvements in cognition and neurobehavioral recovery after TBI remains to be reported.

#### 3.2.2. Botulinum Toxin Type A

Botulinum toxin type A (BoNT-A) is a potent neurotoxin produced by the Gram negative aerobic bacterium Clostridium botulinum. BoNT-A has been used as a pharmacological treatment for the management of spasticity and exerts its effects by binding pre-synaptically to high-affinity recognition sites on the cholinergic nerve terminals. This inhibits the release of acetylcholine, causing temporary neuromuscular blockade and muscle relaxation [28,29]. The effects of BoNT-A are temporary, and neurotransmission slowly resumes as the neuromuscular junction recovers with time [28]. BoNT-A has been used to treat upper limb spasticity after stroke or TBI [29,30,31,32,33]. Clinical studies have shown that a significant reduction in muscle tone is observed as early as one week, with peak effects at 4–6 weeks after injection, waning thereafter [34,35,36].

#### 3.2.3. Hyperbaric Oxygen

Hyperbaric oxygen therapy (HBOT) requires the inhalation of 100% oxygen under a pressure that is greater than 1 atmosphere. Experimental studies of HBOT after TBI demonstrates inhibition of apoptosis, suppression of inflammation, protection of the blood–brain barrier, and promotion of angiogenesis and neurogenesis, with a range of HBOT treatments from 1.5 atmospheres to 3 atmospheres and up to 90 min twice daily for 40 days to 45 min for two sessions [37]. HBOT in humans after brain injury can correct brain anoxia and craniocerebral oedema, reduce intracranial pressure and improve neurological function, prognosis and quality of life in patients [38,39]. In RCTs, HBOT for severe TBI demonstrated higher GCS scores and lower National Institutes of Health Stroke Scale (NIHSS) [40]. The study concluded that higher GCS on admission, tracheotomy status and first HBOT duration were independent prognostic factors in patients with severe TBI [40]. HBOT treatment also led to improved cognitive function in TBI patients [41], however these studies post-TBI are limited and hence further RCTs are required.

#### 3.2.4. Methylphenidate

TBI results in alterations in the chemistry and structure of brain cells and long-term changes in the levels of neurotransmitters. Reduced serotonin and catecholamine are related to TBI-associated neurological comorbidities [42]. Methylphenidate, a psychostimulant which blocks the reuptake of norepinephrine and dopamine into the presynaptic neuron, is used to treat narcolepsy and attention deficit hyperactivity disorder (ADHD) in children [43] but is potentially beneficial against TBI-associated neurological sequelae [44]. Although the exact mechanism is not clear, it is thought to activate the brainstem arousal system, the cortex and subcortical regions such as the thalamus, and produce its stimulant effect on the brain. In a study of ten RCTs, methylphenidate showed significant improvements in enhancing vigilance-associated attention (i.e., selective, sustained and divided attention) in TBI patients, but no significant effects on memory or processing speed were noted [45]. However, most of these previous studies included TBIs with a wide range of severities, age, small samples sizes or use of an open-label design and hence further adequately powered, well-designed, double-blind, placebo-controlled RCTs are required to reach definitive conclusions regarding the use of methylphenidate in TBI [44,46,47,48,49,50,51,52].

#### 3.2.5. NNZ-2566

NNZ-2566 is an analogue of endogenous tripeptide glycine-proline-glutamate with improved stability, and after penetrating ballistic-like brain injury has been shown to be neuroprotective as well as improved motor function and reduced incidence, frequency and duration of post-injury seizures [53,54,55,56]. NZZ-256 exerts anti-inflammatory properties, as it reduces injury-induced increases in pro-inflammatory cytokines, suggesting that NZZ-256 exerts its influence through modulation of the immune response after TBI [57]. Despite the limited number of studies with NNZ-2566 in TBI, Neuren Pharmaceuticals has conducted Phase I and II clinical trials in patients with TBI.

#### 3.2.6. Rivastigmine

Rivastigmine treatment after severe closed head injury reduced cerebral oedema and accelerated motor and cognitive function recovery, effects that were mediated by increased cholinergic activity at both muscarinic and nicotinic receptors [58]. There were some initial Phase I studies of Rivastigmine in TBI showing that it was safe [59,60]; however, an RCT study showed that 17 patients had to withdraw due to adverse events, with 69 patients completing the trial [60]. Clinical interviews, however, failed to show statistically significant positive benefits of Rivastigmine in vigilance tests in the 69 patients that completed the trial [60]. Moreover, a recent Phase III study failed to show any benefits on cognitive function using a Rivastigmine patch in veterans with TBI [61]. 

#### 3.2.7. Tranexamic Acid (TXA)

Uncontrolled hemorrhage after trauma is a cause of early mortality in major trauma, accounting for 30–40% of all deaths. TBI is associated with intracranial bleeds in 25 to 40%, 3 to 12%, and 0.2% of severe, moderate and mild TBI patients, respectively [62]. In addition, one-third of TBI patients suffer from coagulopathy, requiring treatment with antifibrinolytic agents such as TXA [63,64,65]. TXA can therefore reduce bleeding and mortality without adverse effects. TXA is a lysine analogue with anti-fibrinolytic actions and competitively binds to lysine sites of plasminogen and plasmin, inhibiting the binding of plasmin to fibrin and subsequently preventing fibrinolysis [66,67]. 

Despite numerous studies showing some benefits of TXA in TBI, a systematic review of nine RCTs with 14,747 patients found no statistical benefits on mortality or disability after TBI [68]. This was also supported by an Editorial from the Journal “Intensive Care Medicine”, pointing out that the systematic review by Lawati (2021) [68] analyzed all causes of mortality, whereas the CRASH-3 study used “head injury-related death” as a definition to analyze mortality, and this could be prone to information bias [69]. Another study found that prehospital administration of TXA in all TBI patients significantly increased the risk of 30-day mortality rates, with a higher risk in those with severe isolated TBI [70]. Given the current evidence on TXA, further studies are required to clear up whether TXA is beneficial in TBI patients or not.

### 3.3. Ongoing Studies

Several TBI studies are currently ongoing, with 3 studies that are active but not recruiting (Table 2), 37 studies that are currently recruiting (Table 3) and 13 studies are yet to recruit (Table 4). The biggest study planned is the Phase III CRASH-4 trial, with 10,000 participants to be enrolled onto the study, a continuation of the CRASH trials to evaluate the effect of the antifibrinolytic agent, tranexamic acid, in mild head injury in older adults (50 years and older). Other studies of note are NCT04588311, which is a Phase III study that will evaluate the effect of erythropoietin-alpha in preventing mortality and reducing severe disability not only in TBI patients but also other severe trauma patients. NCT03061565 will compare the effects of erythropoietin in reducing mortality over the longer term and hopes to recruit over 600 patients.

Other drugs currently being tested in Phase III studies include Biperiden Lactate to reduce post-TBI epilepsy, dexamethasone to reduce pericontusional oedema, Dalteparin to prevent venous thromboembolism, NT201 to reduce lower limb spasticity after TBI, inhaled nitric oxide to reduce secondary brain damage, Nucleo CMP Forte to protect against glutamate toxicity in children, and the effect of citoflavin to improve cerebral blood flow, restore impaired consciousness and improve cognitive outcomes (Table 3). 

### 3.4. Summary of Drugs to Be Used in Large TBI Studies That Are Currently Active

Examples of some of the largest studies (>500 participants) that are currently registered active on ClinicalTrials.gov (accessed on 3 December 2021) include TXA, erythropoietin, phenytoin sodium, Dalteparin, propranolol, NT201 and Dexamethasone. Section 3.2.7. contains a summary of the results thus far with tranexamic acid in TBI and hence they are not discussed here.

#### 3.4.1. Erythropoietin

Erythropoietin (EPO) is a hemopoietic growth factor with neurocytoprotective effects and is normally produced in the spleen, liver, bone marrow, lung and brain [71]. Although EPO is mainly used in conditions where there is impaired red blood cell production, it is neuroprotective and neuroregenerative by reducing apoptosis, inflammation, oxidative stress and excitotoxicity [72,73]. EPO reduces lesion volume and improves neurobehavioral outcomes after TBI [74]. The clinical benefits of EPO in TBI were later realized but data were conflicting [75,76]. One systematic review suggested that EPO reduced overall mortality and shortened hospitalization time but did not improve neurological outcomes [77]. A more recent systematic review involving seven randomized controlled trials (RCTs) and 1197 TBI patients (611 treated with EPO) found no improvements in acute hospital or short-term mortality but did show a significant improvement in mid-term (6 month) follow-up survival rates [78]. Disappointingly, the study found that EPO was not associated with neurological functional improvements [78].

What is also clear is that there were different EPO treatment doses used across the seven studies, which may account for variability. For example, EPO administration ranged from 500 IU/kg to 40,000 IU, subcutaneously or intravenously injected, some with repeated doses at days and weeks after the initial doses [78]. The majority of the studies administered EPO within 6 h although one study reported up to 24 h. The patient populations were also varied, with some that were reported as severe and some that were moderate in severity. All of these differences could account for the variability in the reported outcomes, and hence uniformity will be important in future studies.

#### 3.4.2. Phenytoin Sodium

Phenytoin is a widely used anti-epileptic drug used to control post-traumatic seizure prophylaxis. The use of anti-epileptic drugs in TBI remains a point of contention. However, it is recognized that post-TBI seizures develop in 12% of severe TBI cases and that phenytoin treatment reduces this to 3.6% [79]. Current guidelines for post-TBI seizures focus on control efficacy, for which phenytoin and Levetiracetam is commonly used. However, phenytoin has several complications which limits its use, including imbalance and dizziness [80]. In a recent systematic review, the authors failed to identify differences between phenytoin and Levetiracetam in any of the outcomes after TBI including early seizures, stating that further well-powered RCTs are required to reach definitive conclusions on the benefits of phenytoin after TBI [81]. In this regard, the MAST trial (NCT04573803) plans to recruit 1649 participants in a Phase III trial to assess phenytoin and Levetiracetam, and aims to answer whether a shorter or a longer course of anti-epileptic drugs prevent further seizures in patients that have started having seizures after TBI, as well as whether a 7-day course of phenytoin and Levetiracetam should be used in patients with serious TBI to prevent seizures from starting. Results from this trial are eagerly awaited and will go some way to answer whether anti-epileptic drugs should be used in TBI patients.

#### 3.4.3. Dalteparin

TBI patients are at high risk of venous thromboembolic events (VTE), defined as either deep vein thrombosis (DVT) or pulmonary embolism (PE). The risk of baseline VTE is approximately 5%, and this increases to 30–60% in patients with TBI [82,83]. Commonly, low molecular weight heparin or unfractionated heparin is used to prevent VTE complications; however, heparin can increase the risk of expansion of intracranial hemorrhages with VTE prophylaxis [82]. In an early study that retrospectively compared VTE rates between Dalteparin and Enoxaparin, no significant differences between the two drugs was observed. The groups treated with Dalteparin tended to be more severe TBI patients and the study concluded that VTE prophylaxis in TBI patients offered high levels of protection against VTE with an extremely low risk of expansion of intracranial hemorrhages [84].

In a recent systematic review that included 21 studies, VTE prophylaxis did not lead to TBI progression and VTE prophylaxis with 24–72 h after TBI was safe in patients with stable injuries [85]. There was also no relationship between hemorrhagic progression and timing of VEP prophylaxis [85]. In an international comparative study in the Netherlands and the USA, almost 80% of trauma patients received VTE prophylaxis, with a greater proportion of patients with VTE afflicted by TBI [86]. VTE occurred in 75–81% of patients despite receiving adequate VTE prophylaxis and within 48 h of injury. This may suggest that patients developing a VTE are at such a high risk that even with chemical prophylaxis treatment, it is not sufficient or is unable to be started early enough post injury to demonstrate a beneficial effect. Future clinical trials will need to address these concerns.

#### 3.4.4. Propanolol

Severe TBI causes a surge in catecholamines such as epinephrine and norepinephrine, and these remain elevated in patients with persistent coma or who are moribund [87]. In those with TBI, plasma norepinephrine levels at 48 hours post injury are predictive of Glasgow Coma Scale (GCS) at 1 week, survival, the number of ventilator days and the length of stay (LOS) [88]. Systemically, this sympathetic surge causes tachycardia, tachypnea, hypertension and hyperpyrexia with associated motor features such as agitation and dystonia [89]. TBI severity also correlates with decreased heart rate, and persistent sympathetic hyperactivity is also associated with increased length of stay in intensive care units, lower cognitive abilities and higher cognitive fatigue [90,91,92].

The use of β-blockade, such as the non-selective β-blocker propranolol in pre-clinical mouse studies, reduced brain oedema, improved neurological outcomes, increased cerebral perfusion and decreased cerebral hypoxia [93,94,95]. Propranolol also reduced the maximum intensity of agitated episodes as well as reduced aggressive behavior months after TBI [96,97]. This was followed up by two parallel clinical trials (NCT01202110, NCT01343329) which used early treatment with propranolol in TBI and reported improvements in short-term endpoints such as heart rate [98]. In further retrospective studies, β-blockade after TBI conveyed 4–23% improved mortality rates [99]. The DASH after TBI trial (NCT01322048), a randomized, double-blind placebo-controlled trial with 48 patients in total (21 with TBI and 26 with placebo), posted some results in 2017 which showed that the study did not detect significant changes in primary outcomes of ventilator-free days. Therefore, further high-quality studies are required to evaluate the potential benefits of propranolol in TBI patients.

#### 3.4.5. NT201

NT201 is botulinum toxin type A (BoNT-A) and was mentioned earlier in this review. Initial studies in patients with brain injury or cerebral palsy offered significant reductions in spasticity in elbow, wrist, fingers and ankle muscles receiving high doses of NT201 [30,31,32,33,34]. The PATTERN study (NCT03992404) is currently still recruiting patients in a study to compare the efficacy and safety of NT201 in the treatment of lower limb spasticity caused by stroke or TBI, with a planned recruitment of 600 participants in a randomized parallel double-blind study. The primary outcomes for this trial will be the modified Ashworth scale–Bohannon (MAS) ankle score at 4–6 weeks with a co-primary outcome of global impression of change scale (GICS) at 4–6 weeks.

#### 3.4.6. Dexamethasone

Cerebral oedema after TBI is a serious complication which corticosteroids, including glucocorticoids, can ameliorate effectively [100]. Dexamethasone is a corticosteroid that acts on the glucocorticoid receptor and efficiently reduces blood–brain barrier (BBB) permeability. However, dexamethasone does not readily cross the BBB and the mechanisms by which dexamethasone is neuroprotective are still not understood. The CRASH study in 2004, which was a multi-center trial to assess the use of corticosteroids acutely after head injury, concluded that the use of steroids in the acute period after injury caused more harm than good and are not recommended in head injury [101]. However, dexamethasone is commonly administered to patients undergoing a variety of neurosurgical procedures [100]. In TBI patients, dexamethasone treatment significantly reduced the volume of vasogenic oedema, decreased the apparent diffusion coefficient and increased fractional anisotropy, suggesting beneficial effects of this drug in TBI patients [102]. However, the study was a prospective observational study with only 30 TBI patients, and so larger studies are required to confirm these results.

The DEXCON-TBI study (NCT04303065) is a multicenter, randomized triple-blind placebo-controlled study that will quantify the effects of the administration of dexamethasone on the prognosis of TBI patients with brain contusions and pericontusional oedema. The primary outcome for the trial is improvements in the Glasgow scale outcome extended (GOSE) measure with a number of secondary outcomes. The study will recruit 600 participants with a short and descending course of dexamethasone. The results of this study are eagerly awaited and will determine if dexamethasone is likely to confer benefits when administered acutely.

## 4. Discussion

This study reports that there were a total of 361 registered clinical trials listed on ClinicalTrials.gov (accessed on 3 December 2021) containing the search terms “traumatic brain injury” OR “TBI” AND “drug” as of 3 December 2021. Of the trials listed, 138 were completed, 116 were withdrawn, suspended or the status was unknown, 46 were still recruiting, 5 were active but not yet recruiting, 13 were not recruiting and 2 were enrolling by invitation. The average number of patients recruited per study was 865 (range 5–200,000) with a median of 50 patients/study. Of the completed studies, 125 different drugs were reported to be evaluated. Only seven drugs appeared in three or more trials, which represented the most promising treatment options for TBI and included amantadine, botulinum toxin type A, hyperbaric oxygen, methylphenidate, NNZ-2566, Rivastigmine and TXA. 

Only 44 of the 138 completed trials posted results on ClinicalTrials.gov (accessed on 3 December 2021) and of these, only 6 studies have been published in a peer reviewed journal. Interestingly, the published studies were only studies that were positive or contained some aspect of positive data that could be reported. All of the other studies with results were largely negative and remained unpublished in peer reviewed journals. Of the studies that posted results, 30 showed no statistical difference between the treatment arm and the placebo/control arm, 2 studies showed worse outcomes and only 1 study demonstrated statistically significant results. In addition, seven of the studies showed better results in the treatment arm but lacked statistical analysis, whilst four studies only included a single arm, presumably due to low study recruitment, and so comparisons were not possible. A significant number of studies also failed to meet their recruitment targets and so a number of studies were underpowered. Surprisingly, some studies recruited more patients than was posted on ClinicalTrials.gov (accessed on 3 December 2021).

Although TXA appears to be the most likely to translate to the clinic, as alluded to in Section 3.2.7 of this review, there is some debate about the benefits of TXA in TBI with a recent systematic review of nine RCTs showing no benefits in mortality or disability [68]. The CRASH-3 study itself enrolled 9,202 head injury patients within 3 h of injury with a GCS of <12 or any intracranial bleeding on computed tomography. The study found no statistical difference in the primary outcome of head injury related death (18.5% with TXA and 19.8% with placebo) (relative risk (RR): 0.94, 95% CI 0.86–1.02). However, there were significant differences in subgroups that were less severely injured (i.e., when those with GCS = 3 or bilateral fixed pupils were excluded (RR: 0.89, 95% CI: 0.80 to 1.00) and in the GCS 9–15 subgroup (RR: 0.78, 95% CI: 0.64 to 0.95), or those that were treated earlier (*p* = 0.005 for time effect) [103]. In these less severely injured patients, it would be expected that they would have the highest mortality benefit and so it may be reasonable to give TXA to this subgroup; however, there is not enough evidence to consider this as standard practice. Furthermore, in the subgroup analysis, there was no indication of the all-cause mortality rate. In contrast, TXA also increased death from all other causes, although the result was not statistically significant (RR: 1.31; 95% CI 0.93–1.85). Moreover, the CRASH-3 trial reported that disability remained unchanged, and so based on the evidence, the trial can be classed as negative, since the primary outcome was not met, and neither was disability affected in favor of TXA. Although it has been claimed that the study is underpowered despite involving over 9000 participants, an official video claiming that TXA “could save 10 s of thousands of lives” are unfounded [104].

Of the studies that were completed and published in peer-reviewed journals, there were other studies that failed to meet the primary outcomes but reported statistically significant data in other outcomes or subgroups. Whether this is helpful in improving the design of future clinical trials remains to be seen. However, these studies highlight several aspects of future clinical trial design that need to be taken into account. Although a variety of drugs are being analyzed in clinical trials in TBI, the variability in the reported study results warrants some discussion and refinement in the design of future clinical trials. One major issue within clinical trials of TBI is that all TBI patients, including severe, moderate and mild TBI, are often included. There are clearly significant pathological differences between severe, moderate and mild TBI and therapeutics will have different levels of benefits depending on injury severity. Even using the GCS to stratify TBI patients, heterogeneity is inevitable, since multiple causes may contribute to the same GCS score including diffuse axonal injury, diffuse swelling, contusion and hematoma.

TBIs are also heterogenous (e.g., extradural/subdural, diffuse axons or focal, etc.), and hence the outcome of potential treatments are governed by multiple factors including injury location, physiology and whether the TBI is associated with extracranial injuries. Approximately a quarter of “mild” head injury patients do not return to work and >80% of patients have associated problems even after one year post-TBI, calling into question the term “mild TBI”. Therefore, future studies should stratify patients carefully prior to enrollment in a clinical trial and target only the same patient severities.

Another key consideration in the design of clinical trials is timing of therapeutic intervention. Some treatments will need to be given as early as possible whilst other treatments can be given later. For example, treatments for oedema and to control bleeding are likely to be required immediately after injury and probably best given during the prehospital period to control these adverse events. Neuroprotective treatments may also need to be given within minutes of the injury, which is difficult, since neurons begin to die within minutes of TBI and delays may mask a real neuroprotective effect of a given drug.

Most acute TBI studies are conducted in intensive care which is a safe and controlled environment. However, secondary injuries are likely to occur during the prehospital period where hypoxia, hypotension and expanding hematoma may cause the greatest amounts of neurological damage and where therapeutic interventions may have the best impact. Hence, future clinicals trials should consider studies in the prehospital environment.

One significant issue is that not all clinical trials are published as manuscripts. Only 6 studies were published from amongst the 44 clinical trials that had results posted on ClinicalTrials.gov (accessed on 3 December 2021). Interestingly, these were either studies that were positive or had some positive data that could be reported. It is therefore a concern that only studies with positive data are being published. All of these clinical trials should be written up and published in peer reviewed journals whether positive or negative. This is especially true since significant amounts of resources, manpower and time have been spent on such clinical trials. The studies themselves have value in the sense that it informs other researchers of particular treatments or the design of better studies to obtain unequivocal data regarding the efficacy of a particular compound. The issue of positive publication bias has been highlighted by many and may result in bias in meta-analyses, leading to distortion of literature and misleading researchers, doctors and even policymakers in their decision making [103,104,105,106]. Therefore, all data, be it positive or negative, should be published as long as the study has been performed rigorously and adheres to high-quality standards in study design and analysis. Not publishing the data may be deemed unethical.

### Future Clinical Trial Designs

New clinical trial designs are being recommended in TBI to enhance therapy development [107]. These include comparative effectiveness studies such as those under way in adults and children, CENTER-TBI and ADAPT studies, respectively [108,109]; adaptive trial design where computer algorithms are used to randomize patients in a blinded fashion, those that show favorable effects, rather than randomizing to the test and placebo/control groups equally [110]; continuous data acquisition using electronic medical records rather than data entry of hourly physiological findings [111,112]; and big-data approaches to identify associations and treatment efficacies [113]. All of these recommendations need to be considered in future clinical trial designs to improve the translation of new therapies into the clinic for the benefit of TBI patients.

## Figures and Tables

**Figure 1 brainsci-12-00527-f001:**
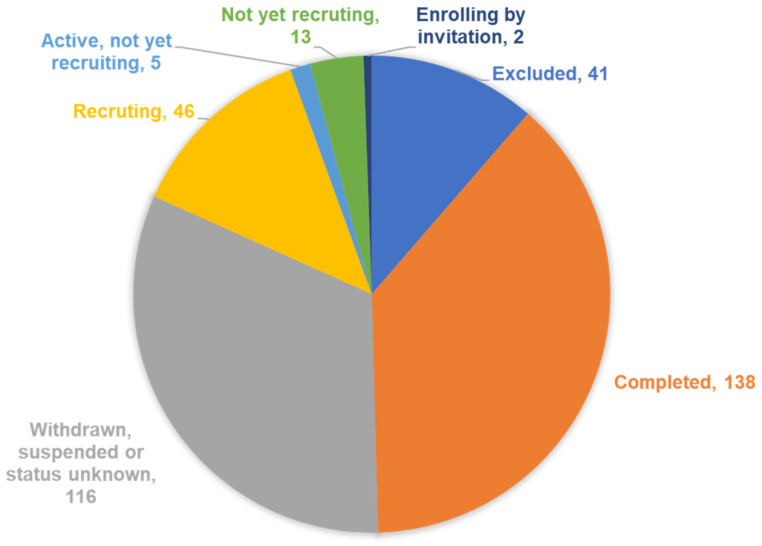
Status of clinical trials in TBI listed on ClinicalTrials.gov as of 3 December 2021.

**Figure 2 brainsci-12-00527-f002:**
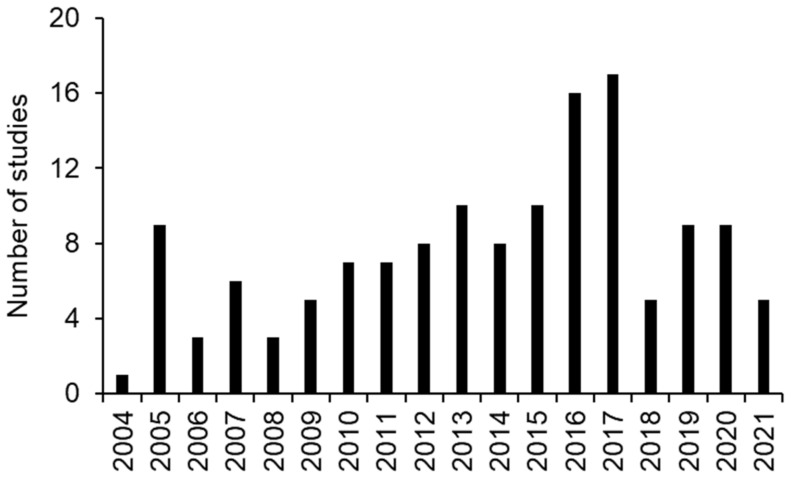
Number of completed clinical trials in TBI in each year from 2002–2021.

**Figure 3 brainsci-12-00527-f003:**
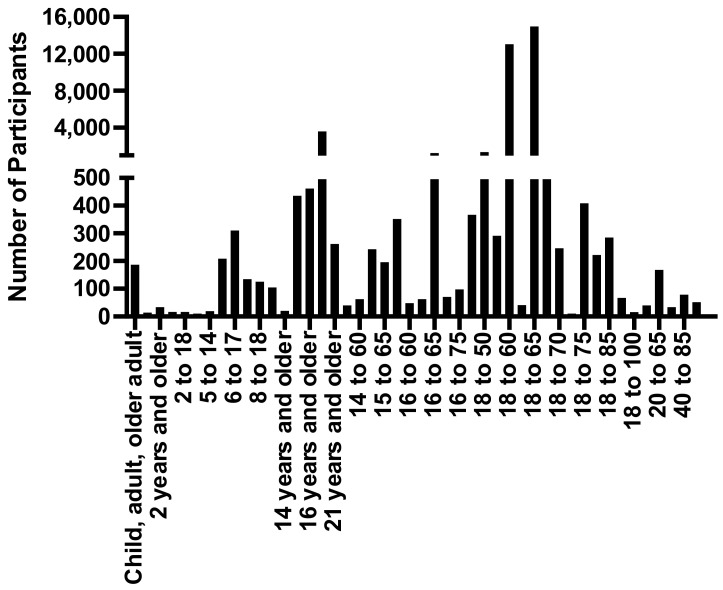
Number of participants for clinical trials in TBI and their age groupings in the completed studies. Note: y axis breaks at 500 participants.

**Figure 4 brainsci-12-00527-f004:**
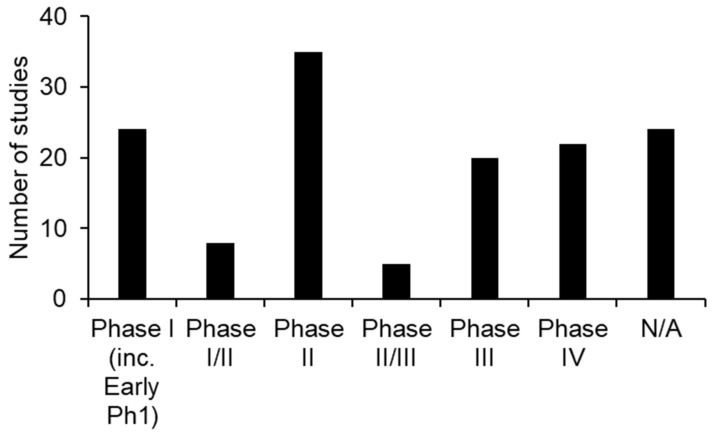
Completed clinical trials in TBI and their study phase/design.

**Figure 5 brainsci-12-00527-f005:**
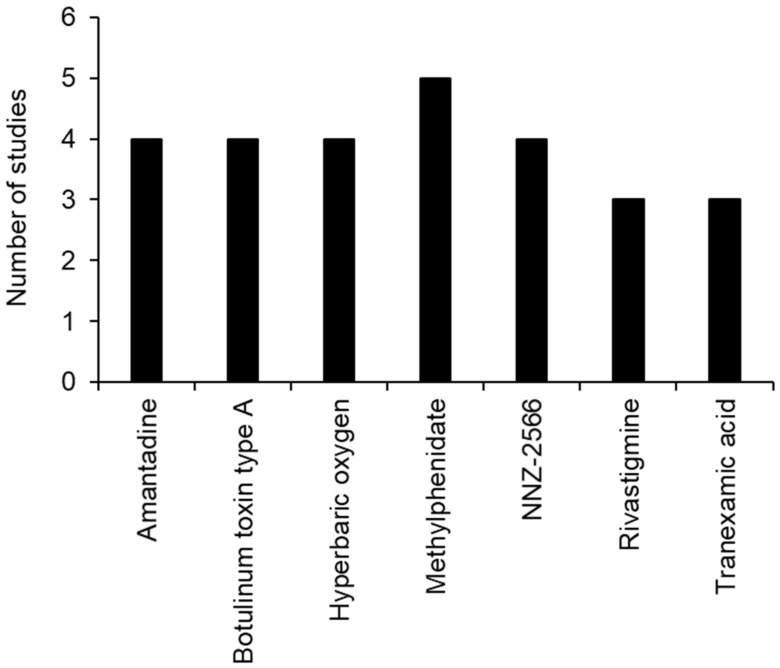
Clinical trials of drugs appearing in three or more registered clinical trials.

**Table 1 brainsci-12-00527-t001:** Completed studies with results summarized.

NCT Number	Intervention	Age (Years)	Enrollment	Primary Outcome	Summary of Results
NCT01990768	Tranexamic acid	15 and older	388	Dichotomized Glasgow Outcome Scale Extended (GOS-E) at 6 months	No statistical difference between 1 g IV TXA followed by 1 g IV maintenance or 2 g TXA IV TXA only and placebo: 967 patients enrolled.
NCT00313716	Recombinant human erythropoietin	15 and older	200	Glasgow Outcome Scale at 6 months	No statistical differences between Epo treatment and placebo. All 200 patients enrolled. Results published.
NCT01322048	IV Propranolol vs. clonidine	16 to 64	63	Ventilator-free days	No statistical change in ventilator-free days using Propanolol and clonidine. Secondary outcomes of plasma norepinephrine levels showed higher levels after treatment: 48 patients enrolled. Results published.
NCT01673828	Allopregnanolone injection	16 to 65	967	Dichotomized Glasgow Outcome Scale Extended (GOS-E) at 6 months	No statistical difference between treatment and placebo in primary outcomes. Only 13 participants enrolled.
NCT00970944	Amantadine Hydrochloride	16 to 65	200	Disability rating score (DRS)	No statistical difference in DRS scores over 6 weeks between treated and placebo. However, DRS score improved faster in the amantadine group in the first 4 weeks, then significantly slower than the normal group: 184 patients enrolled.
NCT01201863	Androgel	16 to 65	41	Restricted functional independence measure (FIM)	No statistical difference between treated and placebo: 46 patients enrolled.
NCT00621751	Carbamazepine	16 to 65	157	Neuropsychiatric Inventory Irritability-Aggression Domains Composite Measure—Observer	No statistical difference between treated and placebo: 70 patients enrolled.
NCT00779324	Amantadine Hydrochloride	16 to 75	52	Proportion of Participants With >2-point increase on Neuropsychiatric Inventory—Irritability Domain rated by Observer day 28	No statistical difference between treated and placebo: 168 patients enrolled.
NCT02957331	Propranolol	18 and older	63	30-day mortality	In the treatment arm, 7.7% of patients died, whilst 33.3% of patients died in the non-treatment arm. No statistical data present: 26 patients enrolled.
NCT01014403	Enoxaparin	18 and older	184	Percentage of participants with worsening TBI hemorrhage	No statistical difference between treated and placebo: 62 patients enrolled.
NCT01847755	Oxygen at 1.5 ATA	18 and older	71	Number of participants with improved cerebral perfusion	Single group allocation and so no comparisons possible: 18 patients enrolled.
NCT00795366	Arginine vasopressin vs. catecholamine	18 and older	60	Time ICP > 20	No statistical difference between treated and standard catecholamine: 96 patients enrolled.
NCT00618436	Levetiracetam vs. phenytoin	18 and older	47	Seizure incidence	No statistical difference between Levetiracetam or phenytoin: 52 patients enrolled.
NCT00233103	Sertraline	18 and older	30	Depression at baseline	No statistical difference between treatment and placebo: 52 patients enrolled.
NCT01368432	Escitalopram	18 and older	30	Montgomery–Asberg depression rating scale (MADRS) at baseline	No statistical difference between treatment and placebo: 16 patients enrolled.
NCT00766038	Recombinant human growth hormone	18 to 50	336	Functional outcome 6 months after injury, as measured by the processing speed index	No statistical difference between treatment and placebo: 63 patients enrolled.
NCT00973674	Premarin IV	18 to 50	256	Percent passing the Galveston orientation amnesia test (GOAT) within 28 days post injury	No statistical difference between treatment and placebo: 50 patients enrolled.
NCT00623506	Pregnenolone	18 to 55	50	Brief assessment of cognition in affective disorders (BAC-A)	No statistical difference between treatment and placebo: 30 patients enrolled.
NCT02225106	Methylphenidate	18 to 55	34	Perceptual organization and processing speed index	Single group allocation and so no comparisons possible: 11 patients enrolled.
NCT00727246	CDP-Choline	18 to 55	13	Cognitive composite score for group of subjects with TBI and healthy controls matched by age, education and treatment group	No statistical difference between treatment and placebo. Low group numbers: 19 patients enrolled.
NCT01856270	Amitriptyline	18 to 60	12,737	Frequency and severity of headaches	No statistical difference between treatment and placebo: 50 patients enrolled.
NCT01342549	Valproate and Naltrexone	18 to 60	15	Time to relapse to heavy drinking	Patients on Naltrexone relapsed to heavy drinking at 15.99 weeks compared to those on Valproate who relapsed after 8.78 weeks: 62 patients enrolled.
NCT02025439	Amantadine	18 to 64	41	Intensity of adverse event	Patients receiving repetitive transcranial magnetic stimulation (rTMS) combined with amantadine (TMS + amantadine) had significantly higher intensity of adverse events. Patient numbers very low: 4 patients enrolled.
NCT01760785	Divalproex sodium	18 to 65	606	Severity of affective lability based on shortened agitated behavior scale	No statistical difference between treatment and placebo: 50 patients enrolled.
NCT02240589	Memantine	18 to 65	238	California Verbal Learning Test—Second Edition (CVLT-II)—Long Delay Free Recall	Memantine group had a higher z-score (−2.000), i.e., worse performance, compared to placebo (−1.375): 11 patients enrolled.
NCT00702364	Atomoxetine	18 to 65	179	CDR power of attention and Stroop test interference T-score	No statistical differences between treatment and placebo: 60 patients enrolled.
NCT02012582	VAS203	18 to 65	100	Effects on intracranial pressure (ICP), cerebral perfusion pressure (CPP), brain metabolism using microdialysis and the therapy intensity level (TIL). Extended Glasgow outcome score (eGOS) at 6 months was exploratory	No statistical differences between treatment and placebo for ICP, CPP and brain metabolism. Scores for eGOS were significantly higher in treated groups: 32 patients enrolled. Results published.
NCT01750268	Topiramate	18 to 65	60	Change in the number of drinking days per week as assessed by the timeline followback (TLFB)	No statistical difference between treatment arm and placebo. Topiramate transiently impaired verbal fluency and working memory. No statistical difference in processing speed, cognitive inhibition and mental flexibility: 32 patients enrolled. Results published.
NCT01611194	Hyperbaric oxygen	18 to 65	50	Summary of treatment-emergent adverse events	Higher incidence of Barotitis media, upper respiratory tract infection and eye disorders in treated group compared to control groups: 71 patients enrolled.
NCT01306968	Hyperbaric oxygen	18 to 65	47	Post-intervention post-concussion symptom scores using RPQ	No statistical differences between treatment and placebo: 79 patients enrolled.
NCT02791945	N-acetylcysteine	18 to 65	32	Change in percent of heavy drinking days per week as assessed by the timeline followback (TFLB)	No statistical differences between treatment and placebo: 30 patients enrolled.
NCT00453921	Methylphenidate	18 to 65	31	Neuropsychological assessment, CVLT-II and CPT, distractibility condition (reaction time)	No statistical differences between treatment and placebo in CVLT-II or CPT: 76 patients enrolled.
NCT01854385	Sumatriptan	18 to 65	16	Change in headache relief	Single arm study so comparisons not possible: 40 patients enrolled.
NCT01249404	Botulinum toxin type A	18 to 80	18	Least squares mean change from baseline to week 4 in the mas score in the gastrocnemius-soleus complex (GSC) (knee extended)	No statistical differences between treatment and placebo at 1000 U of Dysport, but statistically significant at 1500 U Dysport (*p* = 0.0091): 388 patients enrolled.
NCT01313299	Botulinum toxin type A	18 to 80	16	Change from baseline in mas score in the primary targeted muscle group (PTMG)	Treatment 500 and 1000 U of Dysport caused worse outcomes compared to placebo groups: 243 patients enrolled.
NCT00704379	Sertraline	18 to 85	200	Time to onset of diagnostic and statistical manual (DSM) IV defined mood and anxiety disorders associated with TBI	DSM appeared significantly earlier in treated groups (mean 15.78 weeks) compared to 21.42 weeks in placebo treated controls (*p* < 0.05): 94 patients enrolled.
NCT01670526	Rivastigmine transdermal patch	19 to 65	40	Improvements from baseline on the Hopkins verbal learning test-revised (HVLT-R) total recall	No statistical differences between treatment and placebo: 94 patients enrolled.
NCT02270736	NT 201	2 to 17	17	Change from baseline in unstimulated salivary flow rate (uSFR) at week 4	Superiority of NT201 shown with a significant uSFR decrease with NT201 compared to placebo (*p* = 0.0012). Secondary outcome of Carer’s GICS rating was also significant and favored treatment (*p* = 0.032): 256 patients enrolled. Results published.
NCT01322009	Probenecid and N-acetyl cysteine	2 to 18	17	Number of participants who experienced adverse events	None in the treatment arm experienced adverse events; 2 patients in the placebo arm experienced adverse events: 14 patients enrolled.
NCT00957671	Recombinant human growth hormone	21 and older	210	Maximum oxygen uptake at baseline and maximum oxygen uptake after one year	Single arm study so comparisons not possible: 15 patients enrolled.
NCT01336413	Pregnenolone	21 to 55	34	CAPS (Cluster D symptoms)—primary behavioral outcome measure and Tower of London (Subscale Test of BAC)—primary cognitive outcome measure	No significant differences in CAPS between treatment and placebo. Tower of London score slightly better at 4 weeks but worse at 8 weeks compared to placebo: 53 patients enrolled.
NCT01463033	Levetiracetam	6 and older	15	Post-traumatic epilepsy	No significant difference in post-traumatic epilepsy rate in treated versus control arms: 126 patients enrolled. Results published.
NCT02712996	Lisdexamfetamine	6 to 16	150	Assessing severity of symptoms associated with attention-deficit/hyperactivity disorder (ADHD) using the Conners-3 parent form and executive using the behavior rating inventory of executive function (BRIEF)	Better overall scores in the Conners-3 and BRIEF outcomes. No statistical data present: 20 patients enrolled.
NCT01933217	Methylphenidate	6 to 17	300	Parent outcome-Vanderbilt ADHD parent rating scales (VADPRS) and parent outcome-behavior rating inventory of executive functioning (BRIEF)	Slightly better overall scores in VADPRS and BRIEF outcomes. No statistical data present: 26 patients enrolled.

**Table 2 brainsci-12-00527-t002:** Studies active but not recruiting (as of 3 December 2021).

NCT Number	Intervention	Age (Years)	Enrollment	Phase of Trial	Date Last Updated
NCT04833218	Propanolol	18 to 60	90	Early Phase I	6 April 2021
NCT03554265	Somatropin	18 to 70	54	Phase III	30 November 2021
NCT02255799	Donepezil	18 to 60	160	Phase III	26 April 2021

**Table 3 brainsci-12-00527-t003:** Studies active and currently recruiting (as of 3 December 2021).

NCT Number	Intervention	Age (Years)	Enrollment	Phase of Trial	Date Last Updated
NCT04487275	MLC901	15 to 65	80	Phase IV	27 July 2020
NCT05033444	PRV-002	18 to 55	24	Phase I	24 September 2021
NCT01048138	Biperiden Lactate	18 to 75	132	Phase III	18 June 2021
NCT02404779	Cisatracurium besilate	18 and older	34	Phase IV	7 July 2020
NCT04550377	Cannabidiol	18 to 70	120	Phase II	2 June 2021
NCT04303065	Dexamethasone	18 to 85	600	Phase III	23 November 2020
NCT03559114	Dalteparin	18 and older	1100	Phase III	15 June 2021
NCT04489160	C1 Inhibitor	18 to 64	106	Phase II	13 September 2021
NCT04006054	Dexmedetomidine vs. Midazolam	18 to 70	82	Phase IV	2 July 2019
NCT01821690	Buspirone	18 to 70	74	NA	2 April 2021
NCT03061565	Erythropoietin	15 to 75	603	Not specified	25 August 2021
NCT03992404	NT 201	18 to 85	600	Phase III	22 November 2021
NCT04521881	Tranexamic acid	50 and older	10,000	Phase III	20 October 2021
NCT02990091	Omega 3 fatty acid/Safflower seed oil	18 to 55	45	Phase II	23 September 2021
NCT04974060	Remifentanil injection	18 and older	30	NA	29 July 2021
NCT04280965	Quetiapine Fumarate	18 and older	20	Early Phase I	1 March 2021
NCT03982602	Ketogenic diet	18 and older	10	Early Phase I	12 October 2021
NCT03260569	Inhaled nitric oxide	18 and older	38	Phase III	3 November 2020
NCT04527289	Amantadine	18 to 75	50	Phase IV	5 October 2021
NCT04426487	Progesterone	20 to 65	200	Early Phase I	11 June 2020
NCT04508244	Propranolol	18 to 65	771	Phase IV	21 February 2021
NCT04718155	Atorvastatin	18 to 40	30	NA	25 March 2021
NCT04244058	Amantadine + L-DOPA	18 to 75	30	Early Phase I	4 October 2021
NCT05097261	Ketamine	18 and older	100	Phase IV	28 October 2021
NCT03095066	AVP-786	18 to 75	150	Phase II	29 November 2021
NCT04499755	Nucleo CMP Forte	up to 18	100	Phase III	5 August 2020
NCT04558346	Ghrelin (OXE-103)	18 to 60	40	Phase II	28 July 2021
NCT03417492	Sildenafil Citrate	40 to 65	30	Phase I	23 September 2021
NCT04673240	Botulinum toxin type A injection	18 and older	70	Not specified	17 December 2020
NCT02407028	Hyperbaric oxygen	16 to 65 Years	200	Phase II	27 August 2021
NCT03814356	Methylphenidate	18 and older	22	Phase I	2 July 2021
NCT04117672	Salovum (dietary supplement)	10 to 70	20	Phase II	24 March 2021
NCT04588311	Epoetin Alfa 40000 UNT/ML	18 to 75	2500	Phase III	25 August 2021
NCT04631484	Cytoflavin	18 to 60	320	Phase III	18 August 2021
NCT04099667	Rimabotulinumtoxin B	18 to 80	272	Phase II/III	15 July 2021
NCT04710550	F18-3F4AP	18 to 90	66	Phase I	24 February 2021
NCT04744051	Adipose Derived Stem Cell Infusion	18 to 65	20	Phase I	9 February 2021

**Table 4 brainsci-12-00527-t004:** Studies active but not recruiting (as of 3 December 2021).

NCT Number	Intervention	Age (Years)	Enrollment	Phase of Trial	Date Last Updated
NCT05049057	Erenumab	18 to 50	404	Phase II	17 September 2021
NCT04573803	Phenytoin Sodium	10 and older	1649	Phase III	3 November 2020
NCT04945213	Biperiden	18 to 75	312	Phase III	22 October 2021
NCT04003285	Allopregnanolone	21 to 62	132	Phase II	23 September 2021
NCT05058677	Aerosolized 2% lidocaine	up to 16	12	Phase IV	11 October 2021
NCT04400266	B + MEL	18 to 64	10	Phase IV	22 May 2020
NCT04427241	Amantadine Sulfate + Cerebrolysin	19 to 64	12	Phase IV	11 June 2020
NCT04867317	Somatropin	21 to 55	172	Phase III	15 October 2021
NCT05095857	S-ketamine	18 and older	400	Phase IV	27 October 2021
NCT04815967	Rimabotulinumtoxin B	18 to 80	272	Phase II/III	15 July 2021
NCT05008926	Naloxegol	18 and older	370	Phase III	23 August 2021
NCT04387305	Tranexamic acid injection	up to 17	2000	Phase III	12 July 2021
NCT04515420	Noradrenaline	18 to 80	60	Not specified	17 August 2020

## Data Availability

All data generated as part of this study are included in the article.

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
