# Peer review of "Current Clinical Trials in Traumatic Brain Injury"

_brainsci, 2022, doi:10.3390/brainsci12050527_

Round 1
Reviewer 1 Report
The Author documented in compressive way status of traumatic brain injury trials and try to give some tips how improve future clinical trial design.
Introduction,
-short and compressive describe of traumatic brain damage types and mechanisms.
-Author accent problem of clinical treatment of patients after TBI., whereas in my opinion there is need to described also long term disability.
-what is main aim of the review and potentially clinical effect???
Trial
84-93
The trials included in review are from 2004-2021?? What about earliest trials??
-In my opinion, there is no consistency in the name of drugs (105-111)
3. Summary of Evidence for Drugs Appearing in Three or More Studies
Poor describe of results and main topics about drugs…
Some compresive summary of drugs used in trials is need

Author Response
Comment: short and compressive describe of traumatic brain damage types and mechanisms.
Author response: This is deliberately short as there are very many good reviews on TBI in the literature. The point of this article is to present the clinical trials that are currently registered as being performed.
Comment: Author accent problem of clinical treatment of patients after TBI., whereas in my opinion there is need to described also long term disability.
Author response: I have now added a small paragraph to discuss long-term disability in TBI. See Lines 428-441,
Comment: what is main aim of the review and potentially clinical effect???
Author response: This has now been clarified at the end of the last paragraph in the Introduction.
Comment: The trials included in review are from 2004-2021?? What about earliest trials??
Author comment: These are the only ones that are listed in ClinicalTrials.gov. There are no earlier trials from our search ClinicalTrials.gov.
Comment: In my opinion, there is no consistency in the name of drugs (105-111)
Author response: We agree that there is no consistency but these are the drugs that were used in 3 or more trials, which was the criteria for their inclusion here.
Comment: 3. Summary of Evidence for Drugs Appearing in Three or More Studies. Poor describe of results and main topics about drugs…Some compresive summary of drugs used in trials is need.
Author comment: Please see section 3.3 where I have discussed drugs that are currently being used in clinical trials. The drugs appearing in 3 or more studies was supposed to be a brief summary of the studies that have been performed with these drugs. Some of these do re-appear in the new Section 3.3.
Reviewer 2 Report
TBI is one of the leading causes for death and disability around the world and major societal burden, therefore the comprehensive review on clinical trials in TBI by Ahmed, Z is timely and relevant for the field. However, there are some concerns and points of improvement that the author can consider.
- In figure 2. Is there a reason to choose these yearly intervals? It’s confusing because it’s not consistent (5 versus 6 years for some intervals) and the intervals are so crude that it doesn’t give too much information. Dividing the number of studies by year would be more informative.
- In figure 3, why is the status of the completed study unknown? This information should be available somewhere and included in the manuscript.
- Again, for summary concerning the age of patients included in the clinical trials, a more comprehensive and detailed data representation is needed.
- It would be beneficial for the manuscript to have a table with all the ongoing clinical trials as of December 3rd 2021 with all details concerning the study (eg number of patients, phase etc).
- The author chose to discuss in details for the drugs investigated in 3 or more studies. It would be more useful to focus on the drugs currently in trials (they are listed in 2.2 Phase IV Studies) or the drugs that showed success.
Author Response
Comment: In figure 2. Is there a reason to choose these yearly intervals? It’s confusing because it’s not consistent (5 versus 6 years for some intervals) and the intervals are so crude that it doesn’t give too much information. Dividing the number of studies by year would be more informative.
Author comment: Thank you. I have now presented year on year totals from 2004-2021. Makes much more sense!
Comment: In figure 3, why is the status of the completed study unknown? This information should be available somewhere and included in the manuscript.
Author comment: Apologies, we have now completed this.
Comment: Again, for summary concerning the age of patients included in the clinical trials, a more comprehensive and detailed data representation is needed.
Author response: A new figure 3 is included to explain all of the different age ranges used for TBI clinical trials. Hopefully, this range is now better represented.
Comment: It would be beneficial for the manuscript to have a table with all the ongoing clinical trials as of December 3rd 2021 with all details concerning the study (eg number of patients, phase etc).
Author response: I have now included Table 1-3 which detail studies active but not recruiting, studies active and currently recruiting, and studies active but not recruiting.
Comment: The author chose to discuss in details for the drugs investigated in 3 or more studies. It would be more useful to focus on the drugs currently in trials (they are listed in 2.2 Phase IV Studies) or the drugs that showed success.
Author comment: I have now added a section 3.3 on a summary of drugs being used in large studies (>500 participants). Please see lines 269-396.
Round 2
Reviewer 1 Report
Thank You for revision. I don't feel that this topic is important for research in brain area.

Author Response
Comment: Thank You for revision. I don't feel that this topic is important for research in brain area.
Author response: With respect, this article is absolutely required and timely since a review of the current clinical trials being performed is very informative. I have not found anything recent that is in the like of this review. Even though there have been many trials, few have been implemented into the clinical space and there remains a great medical need in TBI. Also, I have tried to incorporate some suggestion from current trial design methodology that must be used to improve future clinical trials.
Reviewer 2 Report
I thank the author for the revision and addressing all my concerns. The manuscript has improved substantially and is suitable for publication in Brain Sciences. I only have several minor comments for the revised manuscript:
- The newly added paragraph Long-term consequences of TBI (line 404) disturbs the flow of the discussion. This paragraph would be more suitable in the introduction, for example after the paragraph on secondary injuries (line 58). The paragraph can also be shortened if needed.
- Line 72: would be better to change "tips" to "guidelines" or similar
- Figure 3 seems to appear twice
Author Response
Comment: I thank the author for the revision and addressing all my concerns. The manuscript has improved substantially and is suitable for publication in Brain Sciences. I only have several minor comments for the revised manuscript.
Author response: Many thanks and thank you for your time in helping me to improve the manuscript. I think your suggestion have been excellent and has made the manuscript a lot better.
Comment: The newly added paragraph Long-term consequences of TBI (line 404) disturbs the flow of the discussion. This paragraph would be more suitable in the introduction, for example after the paragraph on secondary injuries (line 58). The paragraph can also be shortened if needed.
Author response: I agree and have therefore moved the paragraph and have shortened it a little. This is now in Line 68-76. As a result of moving the paragraph, I have also moved the aims of the study to the last paragraph of the Introduction. Lines 77-82.
Comment: Line 72: would be better to change "tips" to "guidelines" or similar.
Author response: Amended to guidelines.
Comment: Figure 3 seems to appear twice
Author response: It’s not that it appears twice, there is a y axis break at 500 participants because of the studies with over 12000 participants does not show studies with low numbers very well. I have added a note in the figure legend to say that there is an axis break.